# Antrodin C Isolated from Antrodia Cinnamomea Induced Apoptosis through ROS/AKT/ERK/P38 Signaling Pathway and Epigenetic Histone Acetylation of TNFα in Colorectal Cancer Cells

**DOI:** 10.3390/antiox12030764

**Published:** 2023-03-21

**Authors:** Yung-Yu Hsieh, Ko-Chao Lee, Kung-Chuan Cheng, Kam-Fai Lee, Ya-Ling Yang, Hsin-Tung Chu, Ting-Wei Lin, Chin-Chu Chen, Meng-Chiao Hsieh, Cheng-Yi Huang, Hsing-Chun Kuo, Chih-Chuan Teng

**Affiliations:** 1Division of Gastroenterology and Hepatology, Department of Internal Medicine, Chang Gung Memorial Hospital, Chiayi 613016, Taiwan; 2College of Medicine, Chang Gung University, Taoyuan 333323, Taiwan; 3Division of Colorectal Surgery, Department of Surgery, Chang Gung Memorial Hospital-Kaohsiung Medical Center, Kaohsiung 833401, Taiwan; 4College of Medicine, Chang Gung University, Kaohsiung 833401, Taiwan; 5Department of Pathology, Chang Gung Memorial Hospital, Chiayi 613016, Taiwan; 6Department of Anesthesiology, Kaohsiung Chang Gung Memorial Hospital and Chang Gung University College of Medicine, Kaohsiung 833401, Taiwan; 7Biotech Research Institute, Grap King Bio Ltd., Taoyuan 325002, Taiwan; 8Division of Colon and Rectal Surgery, Department of Surgery, Chang Gung Memorial Hospital, Chiayi 613016, Taiwan; 9Department of Nursing, Division of Basic Medical Sciences, Chang Gung University of Science and Technology, Chiayi 613016, Taiwan; 10Research Fellow, Chang Gung Memorial Hospital, Chiayi 613016, Taiwan; 11Research Center for Food and Cosmetic Safety, College of Human Ecology, Chang Gung University of Science and Technology, Taoyuan 333324, Taiwan; 12Chronic Diseases and Health Promotion Research Center, Chang Gung University of Science and Technology, Chiayi 613016, Taiwan

**Keywords:** antrodin C, colorectal cancer, apoptosis, TNFα, H3K9K14ac

## Abstract

Background: Antrodin C, a maleimide derivative compound isolated from the ethanol extract of the mycelium of *Antrodia cinnamomea*, is an endemic fungus of Taiwan and a potential chemoprotective agent. However, the molecular mechanisms underlying the mode of action of antrodin C on cancer cells, especially in human colorectal cancer (CRC), remain unclear. Methods: The cell death and ROS of the antrodin-C-treated HCT-116 cells were measured by annexin V–FITC/propidium iodide staining, DCFDA, and Fluo-3 fluorescence staining assays. Moreover, signaling molecules regulating TNFα cell death pathways and ROS/AKT/ERK/P38 pathways were also detected in cells treated with antrodin C by Western blotting and chromatin immunoprecipitation. The effects of antrodin C were determined in HCT-116 cell xenograft animal models in terms of tumor volumes and histopathological evaluation. Results: Treatment with antrodin C triggered the activation of extrinsic apoptosis pathways (TNFα, Bax, caspase-3, and -9), and also suppressed the expression of anti-apoptotic molecules Bcl-2 in HCT-116 cells in a time-dependent manner. Antrodin C also decreased cell proliferation and growth through the inactivation of cyclin D1/cyclin for the arrest of the cell cycle at the G1 phase. The activation of the ROS/AKT/ERK/P38 pathways was involved in antrodin-C-induced transcriptional activation, which implicates the role of the histone H3K9K14ac (Acetyl Lys9/Lys14) of the TNFα promoters. Immunohistochemical analyses revealed that antrodin C treatment significantly induced TNFα levels, whereas it decreased the levels of PCNA, cyclin D1, cyclin E, and MMP-9 in an in vivo xenograft mouse model. Thus, antrodin C induces cell apoptosis via the activation of the ROS/AKT/ERK/P38 signaling modules, indicating a new mechanism for antrodin C to treat CRC in vitro and in vivo.

## 1. Introduction

Carcinoma of the colon or rectum, colorectal cancer (CRC), as the fourth leading cause of cancer-related deaths in the industrialized world, is an aggressive, malignant, and lethal disease with a poor prognosis [1]. Developing more effective drugs to treat CRC from natural products rather than classical chemotherapy is an urgent issue due to the parameters, such as the presence of clinical bowel obstruction or the perforation of the bowel wall, that worsen the overall prognosis [2]. There is growing evidence demonstrating the pivotal role of CRC cells, escaped from apoptosis and altered their tendency for cancer invasion and metastasis in tumor initiation, progression, metastases, and cancer recurrence. CRC cells escape from apoptosis and alter their tendency for cancer invasion and metastasis [3]. The intrinsic apoptosis pathway is initiated by exogenous and endogenous stimuli, for example, chemotherapy and radiotherapy or natural compounds. Moreover, apoptosis plays an important function in the development and elimination of damaged cells. The damage signals, in which the activity of caspase-9 and caspase-3 is increased sequentially by the mitochondria, induce cell apoptosis [4]. An abnormal reactive oxygen species (ROS) is the second messenger and is sensitive to oxidative-damage-mediated cell apoptosis and triggers the extrinsic apoptotic signaling pathways under intrinsic apoptotic signaling stimulation [5]. Receptor-mediated pathways include those activated by death ligands, such as TNFα/TNFR1. Death receptors are cell surface receptors that transmit apoptotic signals initiated by specific ligands and play a central role in instructive apoptosis. Therefore, destroying CRC cells through the induction of cancer cell apoptosis or ROS is a great therapeutic strategy for cancer therapy [6].

*Antrodia camphorata* is an endemic fungus of Taiwan that grows only in the hollow rotten heartwood of old Cinnamomum kanehirae [7]. Its chemical constituents include polysaccharides, triterpenoids, steroids, benzene, and maleic acid derivatives. Studies show that *Antrodia cinnamomea* has extensive bioactivity, including protecting livers [8], abating hepatitis [9], resisting hepatotoxicity, immunomodulation [10], and anticancer [11], anti-inflammation [12], and antioxidant activities [13]. The fruiting body and fermented product of *A. cinnamomea* can be used for treating liver diseases, such as preventing hepatic injury induced by ethanol, carbon tetrachloride, and cytokines, inhibiting hepatitis B virus, improving liver fibrosis, and inhibiting liver cancer cells. Research shows that extracted constituents of *A. cinnamomea* provide resistance to liver cancer, lung cancer, gastric cancer, and leukemia [14]. Developing products for preventing or reducing the damage caused by CRC may expand the health-promoting effects of mushrooms. In this study, we aimed to elucidate the inhibitory role of the alcohol-extracted active constituent of antrodin C from *A. camphorata* mycelia in CRC cell growth and its mechanism.

Studies on the well-known anticancer effect of the antrodin C component, which was purified by ethanol extraction of A. camphorata mycelia, mainly focused on anticancer chemoprevention in lung carcinoma, breast cancer, and head and neck cancer cells [15,16,17]. Most recently, studies revealed that the induction of CRC cell apoptosis involved in the ROS/protein kinase B (AKT)/extracellular signal-regulated kinases (ERK)/P38 signaling pathways induced death activator expression [18,19]. Meanwhile, the correlation mechanisms of Fas Ligand/Fas, tumor necrosis factor α/TNF receptor (TNFα/TNFR), to acetylase H3 Lys K9/K14 histone modification are critical events to induce cell apoptosis through upregulating these death-receptor-related proteins [20]. We investigated the molecular mechanisms by which antrodin C induces cell apoptosis via the ROS and AKT/ERK/P38-pathway-mediated epigenetic histone H3K9K14ac of TNFα in colorectal cancer cells (CRCs).

## 2. Materials and Methods

### 2.1. Antrodia cinnamomea Extracts and Analysis of Antrodin C

Freeze-dried powder of *A. cinnamomea* was obtained from the Biotechnology Center of Grape King Inc., Chung-Li City, Taiwan, Republic of China. *A. cinnamomea* was grown in M25 medium with or without 1% ginger (weight/volume) at 25 °C for 50 days, as previously described [21]. After being cultivated, harvested, lyophilized, and ground to powder, the mycelia were kept in a desiccator at room temperature. The purification procedure of antrodin C from *A. cinnamomea* is described in an earlier publication [21]. The freeze-dried powder of *A. camphorata* mycelia was cultured at 30 °C. On the 10th day of fermentation, ultrasonic oscillation was used to extract 0.5 kg of freeze-dried powder of mycelia with 5 L of alcohol five times. It was then filtered and concentrated by decompression to obtain the alcoholic extract of *A. camphorata* mycelium. *A. camphorata* mycelium was purified to obtain antrodin C. With a flow rate of 1.0 mL/min and a scanning UV wavelength of 254 nm, antrodin C was separated at an approximate ~4.556 min of retention time in the mobile phase, and then, its structure was illuminated by mass spectrometry and NMR data. Ethanol extract of the antrodin C from *A. cinnamomea* mycelia was prepared following the protocol in an earlier published paper [22]. HPLC analysis of antrodin C was executed according to a previous study with minor modifications by COSMOSIL 5C18-AR-II column (250 × 4.6 mm; particle size, 5 μm; Nacalai USA, Inc., Kyoto, Japan). The yield rate of antrodin C (PubChem CID: 641729) in the *A. cinnamomea* mycelia with ethanol extraction was ~1.0 g/kg (Figure 1).

### 2.2. Cell Culture

Colon adenocarcinoma cells DLD-1 (CCL-221) and human colorectal carcinoma HCT-116 (CCL-247) were obtained from the American Type Culture Collection and the Bioresources Collection and Research Center of the Food Industry Research and Development Institute (Hsinchu, Taiwan), respectively. Normal human colonic epithelial cells (HCoEpiCs) from ScienCell Research Laboratories (Carlsbad, CA, USA) were cultured in colonic epithelial cell medium (Cat. SC2951). Cell cultures were performed as previously described [23].

### 2.3. Cell Growth and Cell-Cycle Distribution Analysis

3-(4,5-Di-2-yl)-2,5-ditetrazoliu (MTT) was used for cell viability analysis to evaluate the cytotoxic effect, as previously described [24]. To observe cell morphology characteristic of apoptosis, 4,6-diamidino-2-phenylindole (DAPI)-stained cells were examined by fluorescence microscopy, as previously described [25]. Cell-cycle distribution was analyzed using flow cytometry, as previously described [26].

### 2.4. Measurement of Apoptosis Assay and Reactive Oxygen Species

To examine the morphological characteristics of the cells by the treated drug, cells stained with DAPI were observed under a fluorescence microscope, according to a previous report [24]. Cell apoptosis was measured by co-staining with annexin V–FITC and propidium iodide (PI; Sigma), as previously described [19].

To evaluate the intracellular accumulation of ROS (O_2_-), cells stain were determined by H2DCFDA (2,7-dichlorodihydrofluorescein diacetate). The cells were washed prior to FACS analysis, and Cell Quest software was used (Becton Dickenson, Franklin Lakes, NJ, USA), as previously described [19].

### 2.5. Protein Extraction and Immunoblot Analyses

The cells were lysed with lysis buffer comprising sterile water, cell lysis solution (RIPA Buffer, Thermo Fisher, Waltham, MA, USA), protease inhibitor (Sigma), and phosphatase inhibitor (Sigma). Proteins were separated on sodium dodecyl sulfate (SDS)–polyacrylamide gel by electrophoresis and transferred to the polyvinylidene fluoride member, and protein expression was detected using specific antibodies in the Western-light chemiluminescent detection system (Bio-Rad, Hercules, CA, USA) [27].

### 2.6. Animal Study

The animal experiments were approved by the Institutional Animal Care and Use Committee of Chang Gung Memorial Hospital, Chiayi, Animal Ethics Research Board (IACUC: 2021032210). Male BALB/c-nu nude mice at 4–6 weeks old (18–20 g) were purchased from National Laboratory Animal Center, Taiwan. As previously described [27], subcutaneous injection of the HCT-116 cells (10^6^ cells/0.2 mL) into the flanks of mice was performed, and tumor volumes were monitored and assessed.

### 2.7. Histochemistry and Immunohistochemistry Analysis

For histochemical analyses, after being fixed with 4% formaldehyde, tumor tissues were embedded in paraffin blocks. These tissue slides were stained with hematoxylin and eosin under microscopical examination. In addition to immunohistochemical analysis, every subcutaneous tumor specimen was first blocked, then incubated with monoclonal antiproliferating cell nuclear antigen (PCNA), cyclin D1, cyclin E, matrix metalloproteinase (MMP)-9, and TNFα antibodies (Santa Cruz, CA, USA) overnight. These slides were washed and counterstained with hematoxylin under a microscope. Expression proteins were quantitatively assessed by histochemical analysis, as previously described [28].

### 2.8. Chromatin Immunoprecipitation (ChIP) Analysis

After incubation with 1% formaldehyde at room temperature to crosslink DNA and proteins, cells were treated with 125 mM glycine. An SDS lysis buffer was used to scrape the cells. After that, as previously described, specific antibodies against the histone H3K4me3 and non-immunized rabbit IgG as the “no antibody” negative control were incubated with the cells on a rotary shaker. Specific primers for quantitative PCR were performed to amplify the promoter region of the TNFα genes, and their sequences are shown below: TNFα-118 to-10 bp (5′-CAA GCA TTA TGA GTC TCC GG-3′; 5′-AAG CTG TGT TGA GTC CTG AG-3′). The data were expressed as a percentage of a reference gene, as previously described [19,24,28]. A reference gene is for calculating the percent of input for each ChIP.

### 2.9. Statistical Analysis

All data were expressed as mean ± SD and compared between the groups using Student’s *t*-test or one-way analysis of variance. In all cases, a *p*-value of less than 0.05 was considered statistically significant [19,24].

## 3. Results

### 3.1. Antrodin C Induces Cell Apoptosis of Human Colorectal Cancer Cells

This study developed the *A. camphorata* mycelium fermentation technology with minor modifications (Figure 1) [21,29]. The freeze-dried powder of *A. camphorata* mycelia was cultured at 30 °C. On the 10th day of fermentation, ultrasonic oscillation was used to extract 0.5 kg of freeze-dried powder of mycelia with 5 L of alcohol five times. It was then filtered and concentrated by decompression to obtain the alcoholic extract of *A. camphorata* mycelium. *A. camphorata* mycelium was purified to obtain antrodin C, with PubChem CID of 641729. With a flow rate of 1.0 mL/min and a scanning UV wavelength of 254 nm, antrodin C was shown at an approximate ~4.556 min of retention time in the mobile phase. The effective constituent antrodin C extracted by alcohol was 1.0 mg per gram of *A. camphorata* mycelium (Figure 1). We wanted to determine whether different antrodin C doses of *A. camphorata* mycelium inhibit the growing ability of CRC cells (HCT-116), CRC cells (DLD-1), and human colonic epithelial cells (HCoEpiCs). The results show that the antrodin C dose had an obvious inhibitory effect on the viability of CRC cells HCT116 and DLD-1 (IC50, 50 μM). In contrast, normal epithelial cells, HCoEpiCs, did not show any toxic effect (Figure 2A). Furthermore, DAPI dyeing, which was used to observe the apoptosis of CRC cell nuclear DNA, revealed that antrodin C treatment was capable of inductive cell apoptosis of CRC (Figure 2B).

Moreover, flow cytometry analysis of the cell apoptosis with Annexin-V and PI staining also revealed an obvious effect on apoptotic induction (7%, 14%, and 27%) in HCT-116 cells in a dose-dependent manner (Figure 3). Antrodin C had relatively high bioactivity and selectively inhibited CRC cell growth and induced apoptosis. HCT-116 cells treated with antrodin C dose-dependently increased intracellular ROS (1-fold, 1.4-fold, and 1.4-fold) (Figure 3). The cells revealed the spatial dynamics of extracellular Ca2^+^ signaling (1-fold, 1.1-fold, and 1.2-fold), and exhibited depolarization of mitochondrial potential (ΔΨm) marker JC-1 dye by a decrease in the aggregates red/monomers green fluorescence intensity ratio (0.5, 1.1, and 1.3) compared to the control group without antrodin C treatment, as observed through flow cytometry experiments (Figure 3). According to Figure 3, treatment with antrodin C had a significant G1 arrest stage in the cell cycle. Together, our data indicate that the intracellular ROS-mediated G1 arrest and cell apoptosis participated in antrodin-C-mediated anticancer effects in human HCT-116 cancer cells.

### 3.2. Antrodin-C-Mediated In Vivo Growth Inhibition of Colorectal Cancer Cell Xenograft

Next, we confirmed whether the *A. camphorata* mycelium constituent, antrodin C, dose-dependently inhibited subcutaneous acute growth of an HCT-116 cancer cell xenograft in an in vivo nude mouse model. The cancer cells were transplanted on the left and right parts of the back of BALB/c-nu nude mice for five consecutive days. Our data show that antrodin C treatment significantly reduced the tumor size and weight of the HCT-116 xenograft in nude mice at 18 d (Figure 4), suggesting an inhibitory effect of antrodin C on cancer cell growth, with no toxicity on normal cells during treatment. The result shows that 1.0 mg/kg and 5.0 mg/kg doses inhibited 53% and 45% of tumor volume, respectively, compared to the control group. Regarding the weight of the excised tumors, antrodin C inhibited tumor weight by 65% and 50% (Figure 4). Furthermore, the protein levels of PCNA protein (a marker for cell proliferation), cyclin D1 and cyclin E (key regulators of the cell cycle), and MMP-9 (a key regulator of metastasis in colon cancer), as well as TNFα (a marker for extrinsic cell apoptosis) in nude mice were measured by IHC. Our results show that the expression of PCNA, cyclin D1, and cyclin E was reduced in the antrodin C treatment group (Figure 5). However, treatment with antrodin C increased the level of TNFα. Consistent with the in vitro data, treatment with 1.0 mg/kg and 5.0 mg/kg doses of the *A. camphorata* mycelium constituent, antrodin C, displayed growth inhibition of the CRC cancer cell xenograft.

### 3.3. Activation of the Extrinsic Cell Apoptosis Pathway in HCT-116 Cells by Antrodin C

To further determine the apoptotic mechanism of CRC cells by the A. camphorata mycelium constituent, antrodin C, the levels of cell-death-related proteins, including Bcl-2, Bax, cleaved caspase-3, cleaved caspase-9, and TNFα, were measured in antrodin-C-treated HCT-116 cells using Western blotting. Indeed, treatment with antrodin C upregulated TNFα and downstream proteins of the active form of caspase-3 and -9 and Bax (Figure 6A). The extrinsic signaling pathways focus on the elucidation and analysis of the cell-cycle machinery and signaling pathways that control cell-cycle arrest and apoptosis. In contrast, the anti-apoptotic protein—Bcl-2—and cell-cycle-related proteins (cyclin D1 and cyclin E) were decreased in HCT-116 cells treated with antrodin C (Figure 6A). Furthermore, the phosphorylation of ERK Thr202Tyr204 at 6 h increased in antrodin-C-treated HCT-116 cells in a time-dependent manner. The activation of AKT Thr308 by antrodin C was also observed at an early time point until 6 h and markedly sustained for at least 6 h and 12 h (Figure 6B). Similarly, the phosphorylation of p38 Thr180Thr182 after antrodin C treatment was constitutively activated at 6 h and 12 h (Figure 6B). Indeed, the expression of p-ERK, p-AKT, and p-p38 was upregulated in the antrodin-C-treated group. We investigated whether ROS scavengers and protein kinase inhibitors, such as ERK/AKT/p38, could reverse the induction of apoptosis, are involved in the antrodin-C-mediated epigenetic methylation of FasL and TRAIL promoters. HCT-116 cells were treated with or without the specific ROS scavenger NAC (1 mM), AKT inhibitor wortmannin (10 μM), ERK inhibitor PD98059 (50 μM), or the p38 inhibitor SB203580 (10 μM) in the presence or absence of antrodin C for 24 h. As shown in Table 1, ERK/AKT/p38 inhibitors blocked 50 μM antrodin-C-induced cell death by 10%, 6%, and 5%, respectively. The effects of the ROS scavenger NAC almost blocked antrodin-C-induced cell death by 3%.

### 3.4. Antrodin C Increases the Expression of Tumor Necrosis Factor α via Reactive Oxygen Species and ERK/AKT/p38-Mediated Epigenetic Histone Acetylation

The upregulation of TNFα by multiple signaling molecules induces cell apoptosis through the acetylation of histone H3 [30,31]. Thus, we attempted to study if antrodin-C-induced CRC cell death through ROS/ERK/AKT/p38 is related to the regulation of H3K9K14ac histone modification on the promoter regions of the TNFα gene. The ChIP assay results show that antrodin C treatment increased promoter region acetylation of TNFα genes on histone 3 (H3K9K14ac (Acetyl Lys9/Lys14)), but our results indicate that ROS scavengers and protein kinase inhibitors, such as ERK/AKT/p38, decreased epigenetic histone acetylation (Figure 7). These findings, thus, propose a novel biological property for antrodin C to inhibit the growth inhibition and apoptosis induction of HCT-116 by epigenetic H3K9K14ac histone modification of TNFα.

## 4. Discussion

CRC is one of the most common causes of cancer, and the second leading cause of cancer-related mortality worldwide [2]. Studies have revealed several dysregulated cellular signaling pathways in CRC, leading to the onset of malignant phenotypes [32]. Many drugs for the treatment of CRC have been screened in animal experiments and clinical trials [33]. Epidemiological and pre-clinical data suggest that various natural phytochemicals and dietary foods display chemopreventive properties, and in vitro and animal studies support that the compounds may modulate signaling pathways involved in cell proliferation and apoptosis in cancer cells [34]. Diet- and nutrition-derived bioactive compounds trigger ROS-mediated damage to DNA and apoptotic signals, which are important for the disease outcome in patients during and after therapy [35]. Indeed, ROS functions are involved in cellular apoptosis through the activation of many tyrosine kinases and phosphatases as a class of oxygen-containing and active species [35,36]. Phytochemicals display the property of chemoprevention through the activation of ERK, AKT, and p38 pathways to further induce toxicity in cancer cells [37,38,39,40]. TNF-related apoptosis-inducing ligand (TRAIL) could induce apoptosis upon binding to its death-domain-containing receptor, TRAIL receptor DR5. Several studies have described that the expression of TRAIL receptors is higher in CRC compared with normal colorectal mucosa, and targeted therapy with TRAIL leads to the death of tumor cells [31,41]. The results reported herein reveal that the *A. camphorata* mycelium constituent, antrodin C, exerts antiproliferative action and growth inhibition in CRC cells. The cells exhibit the cytotoxicity and morphological characteristics of cell death and biochemical features that characterize apoptosis, as indicated by the loss of cell viability, chromatin condensation, and cell-cycle retardation at the G1 phase (Figure 2 and Figure 3). Natural chemopreventive agents are relatively safe for normal cells, but they can induce tumor apoptosis through different pathways, such as the cell proliferation pathway (cyclin D1 and c-myc), cell survival pathway (Bcl-2, Bcl-xL, cFLIP, XIAP, and c-IAP1), caspase activation pathway (caspase-8, -3, and -9), tumor suppressor pathway (p53 and p21), death receptor pathway (DR4 and DR5), mitochondrial pathways, and protein kinase pathway (JNK, Akt, and AMPK) [42], as has been demonstrated in several clinical trials [43]. Our results show that the antiproliferative activity of antrodin C in CRC cells was mediated by affecting multiple cancer cell signaling pathways. These cell lines were selected to have a different genetic background, which usually results in a different response to natural chemotherapeutic agents rather than HCoEpiC normal epithelial cells. Studies have shown that *Antrodia cinnamomea* possesses an extensive range of pharmacological activities, including anti-tumor activity, by increasing the expression of p53 and Bax [44]. Further findings showed that antrodin C potently inhibited viability, inducing apoptosis triggered by ROS, and arrested the cell cycle at the G2/M phase via a p53 signaling pathway in lung cancer cells [15]. In colorectal cancer (CRC), the p53 gene is mutated in 43% of tumors, which might even provide activities such as promoting cancer cell proliferation, invasion, and metastasis [45]. In this report, we found that antrodin C exhibited more potent cytotoxicity in HCT-116 cells (wild-type p53) than in DLD-1 cells (mutant p53). Antrodin C treatment of HCT-116 increased the expression of Bax as well as the cleavage of caspase-3 and -9 while reducing the expression of Bcl-2. Therefore, our research aimed to ascertain: firstly, whether antrodin C could inhibit the proliferation of HCT-116 cells by cell-cycle arrest and apoptosis; secondly, whether it is possible to define the precise mechanism of the inhibitory action through ROS/AKT/ERK/P38 signaling pathways and epigenetic histone acetylation of TNFα in colorectal cancer cells; and thirdly, an evaluation in vivo. Although the first antrodin C targeting p53 is in HCT-116, a better understanding of mutant p53 functions will likely pave the way for novel CRC therapies. For the first time, the novel role of antrodin C in inhibiting the in vitro viability of CRC cells in HCT-116 and DLD-1 cells and in an in vivo xenograft mouse model (Figure 4 and Figure 5) was demonstrated. Meanwhile, the tumor volume (tumor size and weight) of the HCT-116 xenograft in nude mice was significantly reduced by i.p. injections with antrodin C (1 or 5 mg/kg/day) (Figure 4). Similarly, protein levels of PCNA, cyclin D1, cyclin E, and MMP-9 decreased, whereas TNFα expression increased in the tumor area of the HCT-116 xenograft in nude mice (Figure 5). According to the mechanism, Figure 6A shows that antrodin C treatment induced both death-receptor-mediated extrinsic apoptosis (caspase-3, caspase-9, Bax, and TNFα) and the mitochondria-mediated apoptotic pathway (Bcl-2) in CRC, followed by altering the assembly of decreased cell-cycle-related proteins such as cyclin D1 and cyclin E [46].

Similarly, dietary phytochemicals, such as capsaicin, isoflavones, catechins, lycopenes, phenethyl isothiocyanate, and piperlongumine, which widely exist in food and nutraceuticals, have also displayed inhibitory effects on cancer cells [34]. These compounds may serve as chemopreventive agents, which have significant anticancer effects through the activation of ROS-derived AKT/ERK/P38 pathways, by regulating cancer growth and apoptotic induction (Figure 6B). The anti-invasive effect of natural phytochemicals was through the arrest of the cell cycle at the G1 phase, and is linked to ROS generation, oxidative stress, and protein kinase C and MAPK signaling pathways [47]. Various epidemiological studies have convincingly argued the role of several dietary phytochemicals with enhanced bioactivity in preventing the occurrence of cancer and its treatment, resulting in increased levels of death receptors belonging to the TNFR gene superfamily, the induction of apoptosis, the arrestment of the cell cycle in the G1 phase, and a reduction in CDKs/cyclins levels [46,48]. By measuring the levels of ROS/p70S6K/NF-κB pathways and p21, the anticancer effect of apoptotic modulating dietary phytochemicals in previous cancer therapy was investigated in human CRC cells [26,49]. For the initiation of ROS generation, ROCK1/LIMK2/Cofilin and PI3K/mTOR/p70S6K pathways have both been studied [19,27,28,50,51]. In addition, other studies have indicated that chemopreventive agents change the organization of the actin cytoskeleton through the regulation of the FAK/AKT/p70S6K/PAK1 signals [50]. Several studies have demonstrated that the active constituent antrodin C can enhance the antioxidant defense system of cells by the nuclear factor erythroid 2–related factor 2 pathway in preventing diabetes-related cardiovascular disease [52]. Several studies have demonstrated the inhibitory effects of antrodin C, a maleimide derivative isolated from *A. camphorata* mycelium, in cancer progression and metastasis, including non-small-cell lung cancer and breast cancer, as well as head and neck squamous cell carcinoma [15,16,17]. Antrodin C potently inhibited the viability of human lung adenocarcinoma cell lines, induced apoptosis triggered by ROS, and arrested the cell cycle at the G2/M phase by suppressing both the Akt/mTOR and AMPK signaling pathways. In addition, antrodin C was able to regulate the epithelial-to-mesenchymal transition and the metastasis of breast cancer cells via the suppression of Smad2/3 and β-catenin signaling pathways. In our study, we demonstrated that the molecular mechanisms by which antrodin-C-induced apoptosis and cell-cycle G1 arrest in HCT-116 cells were associated with the activation of TNFα-mediated extrinsic and mitochondria-mediated apoptotic cell death (Figure 3 and Figure 6). However, antrodin C treatment increases the expression of TNFα through ROS-derived and ERK/AKT/p38 MAPK-signaling-pathway-mediated histone modification of H3K9K14ac, which provides more insight into the mechanisms of anticancer in CRC. Furthermore, antrodin C leads to the inhibition of CRC, and the novel regulatory molecule and its anticancer mechanisms need to be identified by two-dimensional electrophoresis (2-DE)-based proteomic analysis.

The epigenome comprises modifications in the chromatin, including DNA methylation and histone modifications, regulated by post-translational modifications (PTMs), which include the acetylation of H3 and H4, currently under development in anticancer agents, which potentiated the inhibition of the tumor and accompanied by TRAIL activation [53]. In vivo and in vitro studies have indicated that the properties of various dietary polyphenols and phytochemicals modify epigenetic mechanisms that regulate gene expression, resulting in restoring aberrant epigenetic alterations and inducing DNA damage and cell apoptosis upon epigenetic changes [54]. For example, quercetin, curcumin, and epigallocatechin-3-gallate provide a comprehensive epigenetic modification [54]. However, the association of antrodin C with their epigenetic modification and the anticancer effects is still unclear. Interestingly, the histone H3K9K14ac (Acetyl Lys9/Lys14), a critical feature of the active promoter state and the expression of several tumor suppressor genes such as TRAIL, has been found to be regulated in human cancer cells. The expression and function of several tumor suppressor genes or oncogenes, such as TRAIL, are regulated by acetylation [55]. The induction of TRAIL genes is associated with histone H3K9K14 acetylation. Their selective effects on pro-apoptotic genes are involved in the protein-kinase-regulated intrinsic cell death pathways, such as Akt, ERK, and p38 [56,57,58]. Consistently, our study determined that antrodin C therapy significantly increased the activation of Akt, ERK, and p38 signaling pathways and the transcriptional activation of histone H3K9K14 acetylation of the TNFα promoters in HCT-116 cells (Figure 7). Our other data show that antrodin C treatment did not significantly increase the level of histone H3 acetylation on the promoters of TRAIL. Thus, antrodin C may be a novel regulator and a chemopreventive agent of dietary phytochemicals to treat CRC for histone H3 acetylation. The signaling protein kinases, including Akt, ERK, and p38, are exposed by the downstream signaling cascades that lead to the phosphorylation of histones, histone acetyltransferase, and histone deacetylases and serve as an exchange for histone modification to gene transcription and expression. Oxidative stress influences histone modification of H3 and H4 in various ways and, thus, affects a number of post-translational chromatin modifications [35,53]. Further studies are needed to directly determine how antrodin-C-induced chromatin modifications impact the apoptosis of CRC cells and the molecular mechanism of antrodin C induction associated with ROS.

## 5. Conclusions

This study demonstrated, for the first time, the chemoprotective activity of antrodin C from *A. camphorata* mycelium and its underlying mechanism in the treatment of human CRC. Our data demonstrate that the histone H3K9K14ac (Acetyl Lys9/Lys14) modification of the binding TNFα promoters by antrodin C plays a significant role in triggering the death-receptor-mediated extrinsic and mitochondria-mediated apoptotic cell death of HCT-116 cells via the activation of both the ROS-derived and Akt, ERK, and p38 MAPK pathways (Figure 8). Our conclusions associated with antrodin C treatment may be applied to CRC therapies.

## Figures and Tables

**Figure 1 antioxidants-12-00764-f001:**
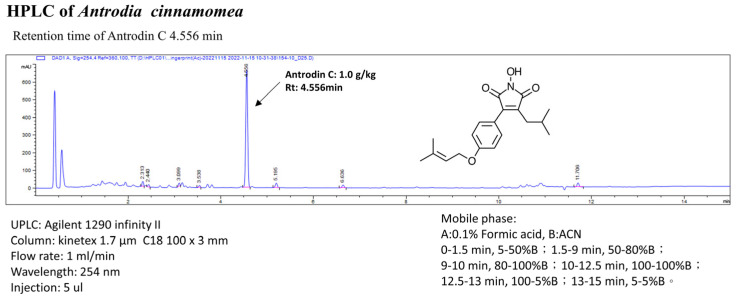
High-performance liquid chromatography analysis of the ethanol Antrodea cinnamomea mycelial extract. The retention time peak of antrodin C from the bioreactor is at 4.556 min (UV detection at 254 nm).

**Figure 2 antioxidants-12-00764-f002:**
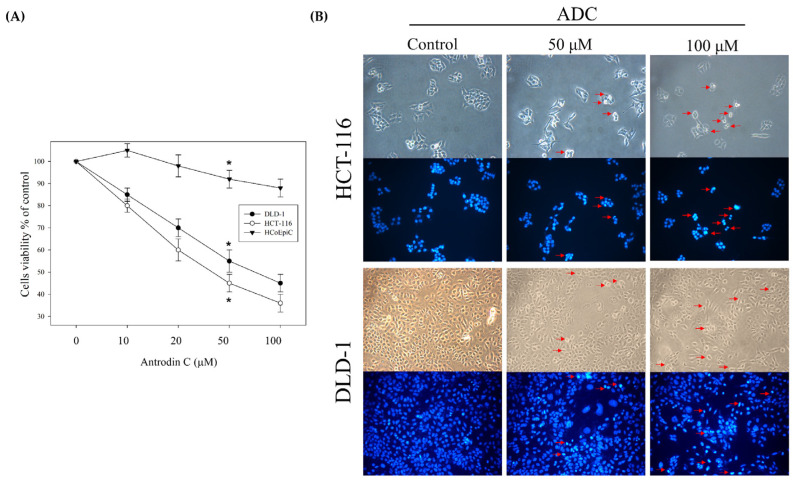
Antrodin C (ADC) decreases cell viability in DLD-1 and HCT-116 cells. (**A**) Cells were treated with either 0.1% DMSO (as control) or ADC for 24 h and subjected to MTT assay for analysis of their cell viability. (**B**) DAPI staining of treated CRC cells was performed to analyze apoptotic cells (red arrowheads) under fluorescence microscopy. The data are expressed as the mean ± SD of three repeats in three independent experiments. * *p* < 0.05, versus the control group.

**Figure 3 antioxidants-12-00764-f003:**
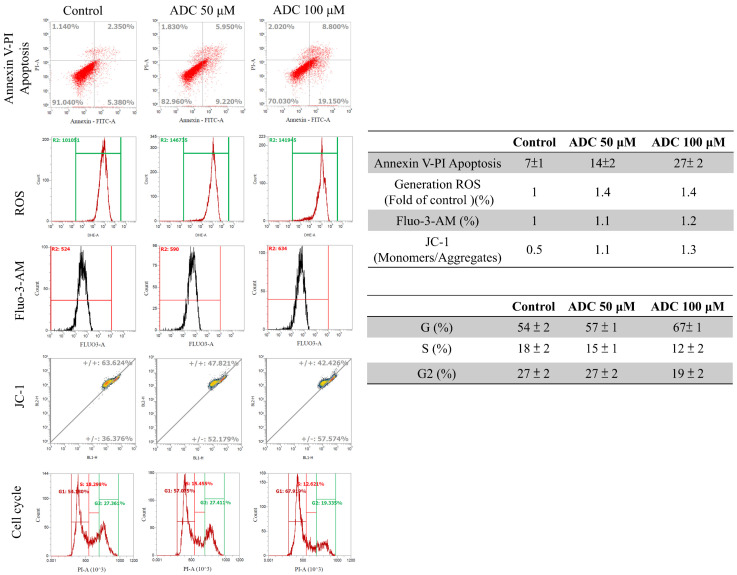
Antrodin C (ADC) induces cell apoptosis and reactive oxygen species (ROS)/Ca2^+^ production as well as decreasing mitochondria potential and cell-cycle G1 arrest in HCT-116 cells. After ADC treatment, the HCT-116 cells were stained with FITC-conjugated annexin-V and propidium iodide (PI) analysis. After individual treatment, the percentage of apoptotic cells was indicated. Measurement of the intracellular ROS/Ca2^+^ of ADC-treated cells was performed by FACS analysis. ROS/Ca2^+^ production is expressed as the fold change of the control group. The ADC-treated cells were stained with JC-1 stain for flow cytometry analysis. A decrease in the green/red fluorescence intensity ratio detected by ADC revealed the membrane-permeant ADC exhibits potential-dependent accumulation in mitochondria. Flow cytometry results show the percentage of cell numbers involved in each cell-cycle phase (G1, S, and G2/M). Data are presented from a list of three independent experiments as mean ± SD.

**Figure 4 antioxidants-12-00764-f004:**
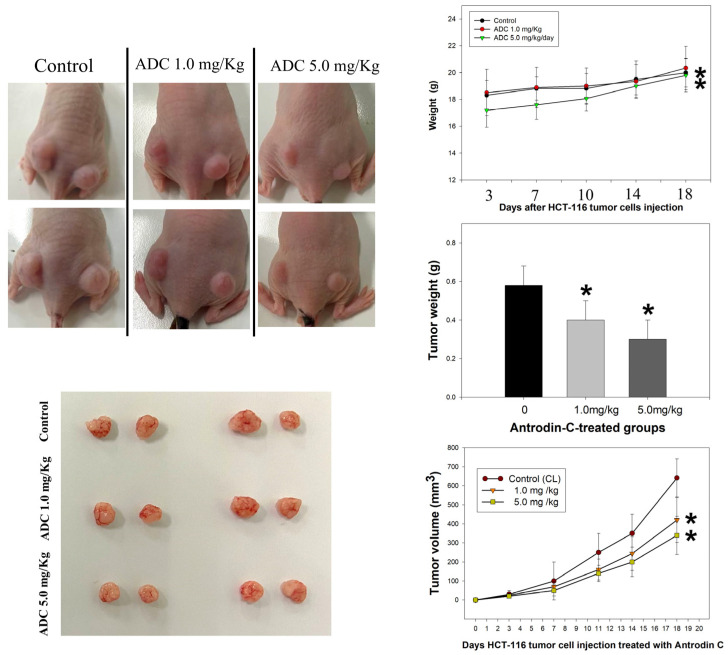
Antrodin C (ADC) repressed the proliferation of CRC cells in an in vivo xenograft mouse model. Subcutaneous injections with 10^6^ cells/mouse for each of the indicated HCT-116 cell lines into nude mice were performed. Representative images of nude mice implanted with colorectal cancer cells, with or without ADC treatment, are shown in left panels. Right panels: after the cancer cells were implanted, the body weight, tumor volume, and tumor weight were measured at intervals of 0, 7, 10, 14, and 18 days with or without ADC treatment. The quantitative data are assayed as mean ± SD. (n = 6/group). * *p* < 0.05.

**Figure 5 antioxidants-12-00764-f005:**
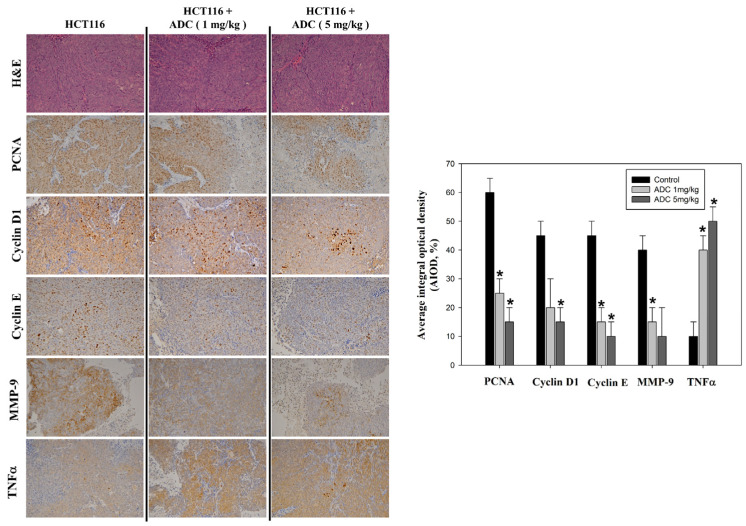
Antrodin C (ADC) varies proliferative and apoptotic protein expression in CRC cells in an in vivo xenograft mouse model. Hematoxylin and eosin staining of the tumors of the nude mice was performed (1st row, upper panel). Protein levels of PCNA (2nd row, upper panel), cyclin D1 (3rd row, upper panel), cyclin E (5th row, upper panel), MMP-9 (6th row, upper panel), and TNFα (7th row, upper panel) were detected by using immunohistochemical analysis. Right panel: By calculating the average of integrated optical density (AIOD), quantitative assay of PCNA, cyclin D1, cyclin E, MMP-9, and TNFα proteins was evaluated. Multiple tumor fields and their positively stained area were evaluated and then evaluated by randomly selecting three observational fields of each section (n = 6/group). The data are expressed as mean ± SD. * *p* < 0.05, magnification × 200.

**Figure 6 antioxidants-12-00764-f006:**
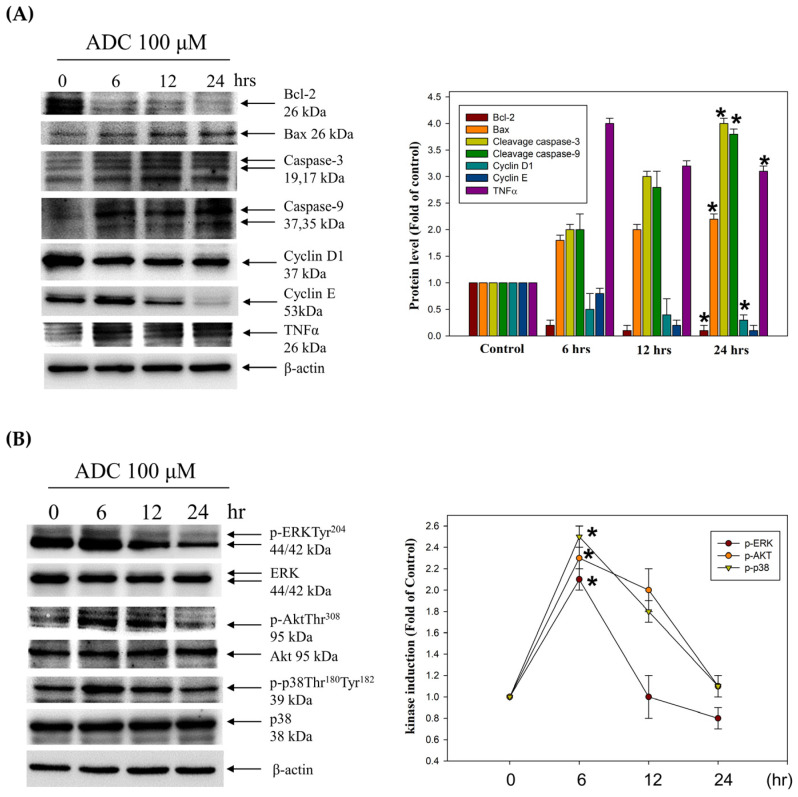
The effect of antrodin C (ADC) on cell death and cell-cycle-related proteins, and ERK/AKT/p38 MAPK pathways in HCT-116 cells. (**A**,**B**) The protein levels of Bcl-2, Bax, caspase-3, caspase-9, cyclin D1, CDK2, cyclin E, TNFα, ERK, AKT, and p38 were determined using Western blotting in ADC-treated HCT-116 cells. While β-actin served as an internal control, densitometric analysis was used to quantify protein levels with the control at 100%. Data are expressed as the mean ± SD of three independent experiments. * *p* < 0.05, compared with the control group.

**Figure 7 antioxidants-12-00764-f007:**
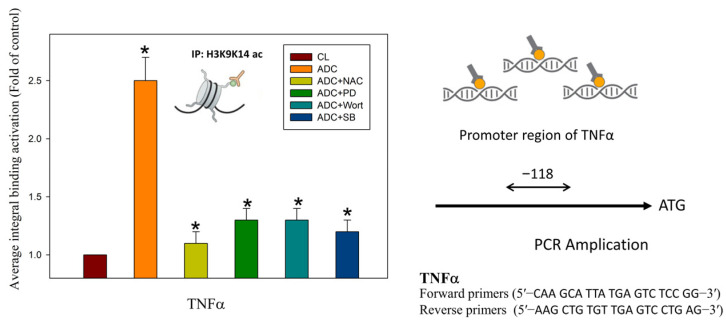
The kinase inhibitors block the binding activities of tumor necrosis factor (TNF) promoter regions, as induced by antrodin C treatment. The chromatin immunoprecipitation assay was performed using antibodies against histone H3K9K14ac (Acetyl Lys9/Lys14) and TNFα promoters (the −118 target sites, as described in the Materials and Methods) in the precipitated DNA, which was amplified by quantitative real-time polymerase chain reaction using specific primer sets. HCT-116 cells were incubated with or without reactive oxygen species (ROS) scavengers, specific ERK1/2 MAPK inhibitor PD98059, PI3K/AKT inhibitor wortmannin, or the p38 MAPK inhibitor SB203580 for 24 h at various concentrations. The data are presented as the mean ± SD of three independent experiments. * *p* < 0.05, as compared to the control group.

**Figure 8 antioxidants-12-00764-f008:**
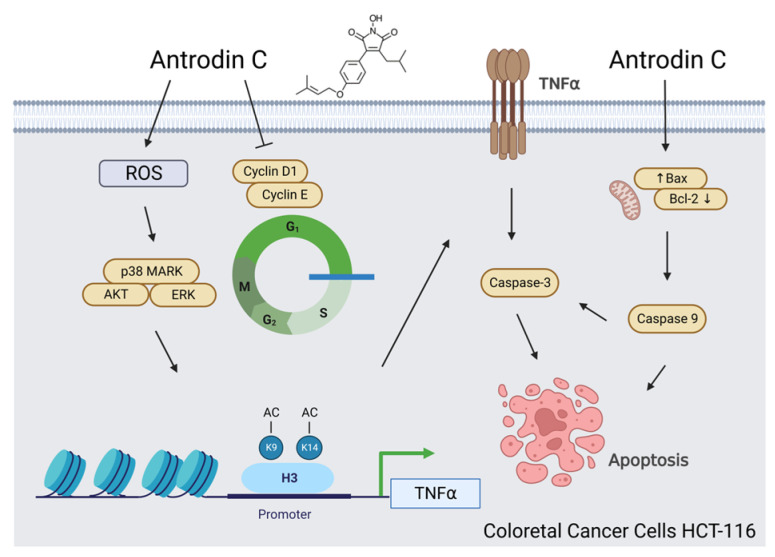
Schematic presentation of the molecular mechanism of antrodin-C-mediated apoptotic induction of colorectal cancer (CRC) cells. Antrodin C treatment increases the expression of death receptor tumor necrosis factor (TNF)α through ROS-derived and ERK/AKT/p38 MAPK-signaling-pathway-mediated histone H3K9K14ac (Acetyl Lys9/Lys14). Activation of TNFα by antrodin C triggers cell apoptosis through the inhibition of Bcl-2 and activation of Bax, caspase-3, and -9. On the other hand, antrodin C treatment also causes cell-cycle arrest at the G1 phase through the downregulation of cyclin D1/cyclin E in HCT-116 cells.

**Table 1 antioxidants-12-00764-t001:** The effects of kinase inhibitors in blocking antrodin-C (ADC)-induced cell death.

	Apoptosis (%)
Control	0
ADC 50 μM	10 ± 3
ADC plus NAC	3 ± 1
ADC plus PD	10 ± 2
ADC plus Wort	6 ± 2
ADC plus SB	5 ± 2

## Data Availability

All relevant data are within the paper.

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
