# Peer review of "Antrodin C Isolated from Antrodia Cinnamomea Induced Apoptosis through ROS/AKT/ERK/P38 Signaling Pathway and Epigenetic Histone Acetylation of TNFα in Colorectal Cancer Cells"

_antioxidants, 2023, doi:10.3390/antiox12030764_

Round 1

Reviewer 1 Report

In this research article the authors investigated the antiproliferative and antioxidant potential of Antrodin C, a compound isolated from Antrodia cinnamomea, a fungus that grows in Taiwan using in vitro and in vivo experiments. The authors performed adequate analysis of the activity of this compound on colon cancer cells and a xenograph mouse model, however several points should be revised.

·       Lines 37-38: The effect of antrodin C and its mechanism were determined in HCT-116 cells xenograft animal models, in terms of tumor volumes and histopathological evaluation.

The measurement of tumor volume and histopathological evaluation are not sufficient to determine the mechanism of action. Please rephrase.

 ·       Lines 84-86.

The sentence is repeated below in lines 91-92.

·       Lines 87-91.

This paragraph makes no sense. Please rephrase.

·       Lines 98-100.

Please avoid describing results in the introduction.

·       Line 108: A. cinnamomea is described in an earlier publication.

Please provide the appropriate reference.

·       Line 144. 2x RIPA lysis buffer.

Provide more details.

·       Lines 182-191.

This paragraph should be moved to Materials and Methods.

·       Line 217: Figure 2. Antrodin C (ADC) decreases cell viability in DLD-1, HCT-116, and HCoEpiC cells.

This is not the case for HCoEpiC cells as mentioned in line 195.

·       Line 210: our data showed a novel anticancer effect.

This statement is exaggerated. Please rephrase.

·       Line 304.

All protein kinase inhibitors exerted the same effect in blocking the binding activities of the TNF promoter regions, as induced by antrodin C treatment. The authors should comment on the lack of specificity detected.

·       Line 332.

MAPK is ERK.

·       Line 341: Figures. 1, 2, and 3.

Figure 1?

·       Lines 380-383.

This is a confusing paragraph. Please rephrase.

·       Lines 422-423

Further validation should be performed in order to support this claim. Please rephrase.

Author Response

Reviewer 1 Comments and Suggestions for Authors

In this research article the authors investigated the antiproliferative and antioxidant potential of Antrodin C, a compound isolated from Antrodia cinnamomea, a fungus that grows in Taiwan using in vitro and in vivo experiments. The authors performed adequate analysis of the activity of this compound on colon cancer cells and a xenograph mouse model, however several points should be revised.

Lines 37-38: The effect of antrodin C and its mechanism were determined in HCT-116 cells xenograft animal models, in terms of tumor volumes and histopathological evaluation. The measurement of tumor volume and histopathological evaluation are not sufficient to determine the mechanism of action. Please rephrase.

Response: Thanks for the comments. The sentence was rephrased in the page 1, line 39-40.

Lines 84-86. The sentence is repeated below in lines 91-92.

Lines 87-91. This paragraph makes no sense. Please rephrase.

Lines 98-100. Please avoid describing results in the introduction.

Line 108: A. cinnamomea is described in an earlier publication.

Please provide the appropriate reference.

Line 144. 2x RIPA lysis buffer.

Provide more details.

Response: Thanks for the comments. These were revised in the page 2, line 88-91, page 2, line 97-99, page 3, line 107, page 3, line 145.

Lines 182-191.

This paragraph should be moved to Materials and Methods.

Line 217: Figure 2. Antrodin C (ADC) decreases cell viability in DLD-1, HCT-116, and HCoEpiC cells.

This is not the case for HCoEpiC cells as mentioned in line 195.

Line 210: our data showed a novel anticancer effect.

This statement is exaggerated. Please rephrase.

Line 304.

All protein kinase inhibitors exerted the same effect in blocking the binding activities of the TNF promoter regions, as induced by antrodin C treatment. The authors should comment on the lack of specificity detected.

Line 332.

MAPK is ERK.

Line 341: Figures. 1, 2, and 3.

Figure 1?

Lines 380-383.

This is a confusing paragraph. Please rephrase.

Lines 422-423

Further validation should be performed in order to support this claim. Please rephrase.

Response: Thanks for the comments. These were revised in the page 3, line 107-120, page 5, line 212-214, page 9, line 313-317, page 10, line 343-344, page 10, line 353, page 12, line 419-422, page 12, line 462-463.

Reviewer 2 Report

Hsieh et al. investigated that antrodin C induced apoptosis through ROS/AKT/ERK/p38 pathway and histone acetylation of the promoter of TNF-alfa. However, there are still several points that the authors should revise their manuscript before the acceptance for “Antioxidants”.

1) As the authors mentioned in the Introduction section, several papers about the effects of antordin C on other types of cancer cells have been already published. Therefore, the authors should discuss both similarities and differences of the effects of antrodin C among the types of cancers in the Discussion section in detail.

2) In Figure 2A, the authors used two colon cancer cell lines, HCT-116 and DLD-1. HCT-116 cells have wild type of p53, but DLD-1 cells have mutant type of p53. As you know that p53 regulates the induction of both apoptosis and G1 cell cycle arrest. Therefore, the authors should investigate the effects on DLD-1 cells and demonstrate whether p53 is involved or not in the effects of antrodin C.

3) In Figure 2A, normal epithelial cells HCoEpiC did not show any effects of antrodi C. It is favorable point from the view of side effects, but the authors should investigate why or how normal epithelial cells are resistant to antordin C.

4) In Figure 3, antrodin C induced both G1 arrest and apoptosis with the generation of ROS, and in Figure 6B, antrodin C activated ERK/AKT/p38 pathway.

The authors should investigate whether ROS scavengers and protein kisase inhibitors such as ERK/AKT/p38 could reverse the induction of G1 arrest and/or apoptosis as same as they did in Figure 7.

5) In Discussion section, the authors described that TRAIL gene is regulated histone H3K9K14 acetylation, and demonstrated antrodin C induced histone H3K9K14 acetylation of TNF-alfa promoter. The authors should show whether antrodin C induced the expression of TRAIL and histone H3K9K14 acetylation of TRAIL promoter.

Author Response

Reviewer 2 Comments and Suggestions for Authors
Hsieh et al. investigated that antrodin C induced apoptosis through ROS/AKT/ERK/p38 pathway and histone acetylation of the promoter of TNF-alfa. However, there are still several points that the authors should revise their manuscript before the acceptance for “Antioxidants”.
 1) As the authors mentioned in the Introduction section, several papers about the effects of antordin C on other types of cancer cells have been already published. Therefore, the authors should discuss both similarities and differences of the effects of antrodin C among the types of cancers in the Discussion section in detail.
Response: We appreciate this remark of reviewer #2’s comment and respond to the comment in the Discussion section.  
Natural A. camphorata mycelium, antrodin C, have been recognized for their anti-cancer effects on many cancer types, including non-small cell lung cancer and breast cancer as well as Head and neck squamous cell carcinoma through Akt/mTOR, TGF-β1, Smad2/3, β-catenin and signaling pathways as well as dysregulatory autophagic signaling [15-17]. However, whether the effective constituent antrodin C isolated from the cultured mycelia of A. camphorata has the same ability of anticancer and signaling activation remains unclear. We demonstrated that antrodin C-induced apoptosis in CRC HCT-116 cells was associated with the activation of TNFα death-receptor-mediated extrinsic and the mitochondria-mediated apoptotic cells death (Figure 6).
These were revised in the discussion section, page 11-12, line 409-417.
2) In Figure 2A, the authors used two colon cancer cell lines, HCT-116 and DLD-1. HCT-116 cells have wild type of p53, but DLD-1 cells have mutant type of p53. As you know that p53 regulates the induction of both apoptosis and G1 cell cycle arrest. Therefore, the authors should investigate the effects on DLD-1 cells and demonstrate whether p53 is involved or not in the effects of antrodin C.
 Response: We appreciate this remark of comment.
Studies have shown that Antrodia cinnamomea possesses an extensive range of pharmacological activities, including anti-tumor activity by increasing expression of p53 and Bax [44]. Further findings showed that antrodin C potently inhibited the viability, induced apoptosis triggered by ROS, and arrested the cell cycle at the G2/M phase via a p53 signaling pathway in lung cancer cells [15]. In colorectal cancer (CRC), the p53 gene is mutated in 43% of tumors that even might provide activities such as promoting cancer cell proliferation, invasion, and metastasis [45]. In this report, we found that antrodin C exhibited more potent cytotoxicity in HCT-116 cells (wild-type p53) than in DLD-1 cells (mutant p53). Antrodin C treatment of HCT-116 increased expression of Bax, as well as cleavage of caspase-3 and -9, while reducing expression of Bcl-2. Therefore, our research aimed to ascertain: firstly, whether antrodin C could inhibit the proliferation of HCT-116 cells by cell cycle arrest and apoptosis; secondly, whether it is possible to define the precise mechanism of the inhibitory action through ROS/AKT/ERK/P38 signaling pathway and epigenetic histone acetylation of TNFα in colorectal cancer cells; and thirdly, to evaluate in vivo. Although the first antrodin C targeting p53 are in HCT-116, a better understanding of mutant p53 functions will likely pave the way for novel CRC therapies.
1.    Scientific reports, 9(1), 5145. 
2.    Molecules (Basel, Switzerland) vol. 24,5 993. 12 Mar. 2019,  
3.    Cancers (Basel). 2021 Apr 28;13(9):2125.
This was discussed and revised in the manuscript, page 11, line 361-377.
3) In Figure 2A, normal epithelial cells HCoEpiC did not show any effects of antrodin C. It is favorable point from the view of side effects, but the authors should investigate why or how normal epithelial cells are resistant to antordin C.
Response: We appreciate this remark of reviewer’s comment. 
Nature chemopreventive agents were relatively safe for normal cells, but it can induce tumor apoptosis by different pathways, such as cell proliferation pathway (cyclin D1, c-myc), cell survival pathway (Bcl-2, Bcl-xL, cFLIP, XIAP, c-IAP1), caspase activation pathway (caspase-8, 3, 9), tumor suppressor pathway (p53, p21) death receptor pathway (DR4, DR5), mitochondrial pathways, and protein kinase pathway (JNK, Akt, and AMPK) [42], as it has demonstrated in several clinical trials [43]. Our results showed that the anti-proliferative activity of antrodin C in CRC cells was mediated through affecting multiple cancer cell signaling pathways. These cell lines were selected to have a different genetic background which usually results in different response to nature chemotherapeutic agents rather than HCoEpiC normal epithelial cells.
1.    AAPS J 11, 495–510 (2009)
2.    AAPS J. 15, 195–218
This was discussed and revised in the discussion of manuscript, page 10-11, line 352-361.
4) In Figure 3, antrodin C induced both G1 arrest and apoptosis with the generation of ROS, and in Figure 6B, antrodin C activated ERK/AKT/p38 pathway.
The authors should investigate whether ROS scavengers and protein kinase inhibitors such as ERK/AKT/p38 could reverse the induction of G1 arrest and/or apoptosis as same as they did in Figure 7.
Response: We appreciate this remark of comment.
To investigate whether ROS scavengers and protein kisase inhibitors such as ERK/AKT/p38 could reverse the induction of apoptosis. Involvement of the ROS-derived and AKT/FAK/PAK1 signals in antrodin C–mediated epigenetic methylation of FasL and TRAIL promoter. HCT-116 cells were treated with or without the specific ROS scavenger NAC (1 mM), AKT inhibitor wortmannin (10 μM), ERK inhibitor PD98059 (50 μM), or the p38 inhibitor SB203580 (10 μM) in the presence or absence of antrodin C (50 μM) for 24 h. As shown in reply 1, ERK/AKT/p38 inhibitors not most blocked 50 μM antrodin C-induced cell death by 10%, 6% and 5%, respectively. The effects of ROS scavenger NAC almost blocked antrodin C-induced cell death by 3% (Table 1).
This was described in the manuscript, page 8, line 288-297, and table 1.

5) In Discussion section, the authors described that TRAIL gene is regulated histone H3K9K14 acetylation, and demonstrated antrodin C induced histone H3K9K14 acetylation of TNF-alfa promoter. The authors should show whether antrodin C induced the expression of TRAIL and histone H3K9K14 acetylation of TRAIL promoter.
Response: We appreciate this remark of comment.
For instance, histone H3 acetylation on K9 and K14 (histone H3K9K14ac) signifies the well-established markers of active gene transcription. To determine if histone H3K9K14ac participated in the antrodin C -upregulated genes expression (TRAIL), we also checked the ChIP level of histone H3K9K14ac (Acetyl Lys9/Lys14) in HCT-116 cells with antrodin C. Our data showed that antrodin C treatment not significantly increased the level of histone H3 acetylation on the promoters of TRAIL at 24 h. 
This was described in the manuscript, page 12, line 438-440.
TRAIL
5′ - TGCATGGATCCTGA GGGCAAGG -3′
5′ -TTGAACCTGCAACTGTCCCTCCC-3′

Reviewer 3 Report

The paper from Yung-Yu Hsieh et al.,  focused on the molecular mechanisms underlying the mode of action of antrodin C in human colorectal cancer (CRC). The paper was well written and was developed both in vitro and in vivo. I request that only a few minor points be explored

1. Fig 6 Caspase 3 is too much fragmented and the distance between the total form and the cleaved is too short. By my experience they are not so close, please attach a new gel with the prestained molecular weight visible , to check the real caspase total dna cleaved migration. The same problem is preset for caspase 9, again show the prestained standard close to the band

2.Fig6 (B). Why does Total ERK1/2 only recognize one band? But then two phosphorylated bands are shown. If only one band is expressed, either 42 or 44KDa then I expect the corresponding phosphorylation.

3.  Could you please show by WB  that the treatment with  antrodin C increase acetylation of H3 and that using a specific si RNA for AcH3 the status of ERK/AKT/p38?

Author Response

Reviewer 3 Comments and Suggestions for Authors

The paper from Yung-Yu Hsieh et al., focused on the molecular mechanisms underlying the mode of action of antrodin C in human colorectal cancer (CRC). The paper was well written and was developed both in vitro and in vivo. I request that only a few minor points be explored

  1. Fig 6 Caspase 3 is too much fragmented and the distance between the total form and the cleaved is too short. By my experience they are not so close, please attach a new gel with the prestained molecular weight visible, to check the real caspase total dna cleaved migration. The same problem is preset for caspase 9, again show the prestained standard close to the band.

Response: We appreciate this remark of comment and respond to the comment as the use of cleaved Caspase-3 (Asp175) and cleaved Caspase-9 (Asp315) antibodies to detects endogenous levels of the large fragment (19/17 kDa) of activated caspase-3 and the large fragment (37/35 kDa) of activated caspase-9 antibodies. Rabbit polyclonal antibodies were purchased from Santa Cruz Biotechnology (Cell Signaling Technology, USA, 1:1000, #9661 and #9505). We had a new gel with the prestained molecular weight visible to check the real caspases (Replay 1).

These are revised in the fig. 6A.

  1. Fig6 (B). Why does Total ERK1/2 only recognize one band? But then two phosphorylated bands are shown. If only one band is expressed, either 42 or 44 KDa then I expect the corresponding phosphorylation.

Response: We appreciate this remark of comment. The use of mouse monoclonal ERK 1/2 Antibody (C-9) and mouse monoclonal p-ERK antibodies were purchased from Santa Cruz Biotechnology (Santa Cruz, CA, USA, 1:1000, sc-514302 and sc-7383). p-ERK (E-4) is recommended for detection of ERK 1 (p44) phosphorylated at Tyr 204 and phosphorylated ERK 2 (p42) of human origin by Western Blotting. Fig.6B western blot analysis of total ERK 1 (p44) and ERK 2 (p42) expression, correspondingly, both phosphorylated ERK1 (p44) ERK 2 (p42) were indicated in HCT-116 whole cell lysates (Replay 2).

These are revised in the fig. 6B.

  1. Could you please show by WB that the treatment with antrodin C increase acetylation of H3 and that using a specific si RNA for AcH3 the status of ERK/AKT/p38?

Response: We appreciate this remark of reviewer #3’s comment and respond to the comment as follows:

AKT1 and p38α and ERK2 by siRNA were determined for histone 3 (H3K9K14ac (Acetyl Lys9/Lys14)) cell protein markers expression by western blot.

Our data showed that the protein phosphorylation level of AKT1 and p38α and ERK2 were downregulated by siRNA in HCT-116 cells (Reply 3). AKT1 and p38α and ERK2 (sc-29195, sc-29433 and sc-353352) or scrambled siRNA (sc-37007) purchased from Santa Cruz Biotechnology. Subsequently, we studied the effects of altered H3K9K14ac totally protein expression levels on biologic processes in human HCT-116.

However, we have shown that antrodin C not be marked increase acetylation of H3 in HCT-116 cells with or without antrodin C treatment. And then our study determined that AKT1 and p38α and ERK2 by siRNA inhibited acetylation of H3 in HCT-116 cells with antrodin C treatment, respectively cells (Reply 3). I wonder whether AKT1 and p38α and ERK2 alone by siRNA regulated acetylation of H3 may be better to determine the status in HCT-116 cells (Replay 3).

Round 2

Reviewer 1 Report

The authors have addressed most of my comments satisfactorily. The resubmitted manuscript is suitable for publication.

Author Response

Reviewer 1 Comments and Suggestions for Authors

Comments and Suggestions for Authors

The authors have addressed most of my comments satisfactorily. The resubmitted manuscript is suitable for publication.

Response: We appreciate this remark of the reviewer #1’s comment.

Reviewer 2 Report

Hsieh et al. investigated that antrodin C induced apoptosis through ROS/AKT/ERK/p38 pathway and histone acetylation of the promoter of TNF-alfa.  Their revised manuscript has not revised completely in accordance with the reviewers’ comments.  Therefore, there are still several points that the authors should revise their manuscript before the acceptance for “Antioxidants”.

1) In their answer to the comment 1 of reviewer #2, the authors described that their research question is whether the effective constituents antrodin C isolated from the cultured mycelia of A. camphorate has the same ability of anticancer and signaling activation remains unclear. However, it seems to be a weak motive, and if so, the research question and their own answer/data are not indicated in their whole manuscript from the points of the view. Therefore, the authors should describe their fundamental research question for this investigation, as previously pointed out as “the authors should discuss both similarities and differences of the effects of antrodin C”. 

2) Their answer to the comment 3 of reviewer #2 is insufficient, but their results in their answer to the comment 4 of reviewer #2, which demonstrated the importance of ROS production by antrodin C, is quite meaningful. Therefore, the authors should investigate whether antrodin C induce ROS production in normal epithelial cells HCoEpiC, and compare with that in colon cancer cells HCT-116.

Author Response

Mar 10, 2023

Ms. Ella Chen
Assistant Editor
Antioxidants Editorial Office

Manuscript ID: antioxidants-2248688 - Minor Revisions

Dear Associate Editor,

Enclosed please find one revised version entitled: Antrodin C isolated from Antrodia cinnamomea Induced Apoptosis through ROS/AKT/ERK/P38 signaling pathway and epigenetic histone acetylation of TNFα in Colorectal Cancer Cells, which we would like to submit for publication in cells.

This revised version has been carefully corrected according editor and referee’s reports point-by-point. We appreciate these valuable comments to strengthen our presentation. Please inform me if any revision is needed. The file marked change in blue color.

Furthermore, I would verify that no part of the manuscript is under consideration for publication elsewhere and it will not submit elsewhere if accepted by cells and not before the Editorial Office has reached a decision.

Sincerely yours,

Hsing-Chun Kuo, Ph.D.

Professor

Department of Nursing,

Chang Gung University of Science Technology,

Chia-Yi Campus,

Taiwan.                                                                                    E-mail: [email protected]

TEL: +886-5-3628800

FAX: +886-5-3628866;

Reviewer 2 Comments and Suggestions for Authors

Hsieh et al. investigated that antrodin C induced apoptosis through ROS/AKT/ERK/p38 pathway and histone acetylation of the promoter of TNF-alfa.  Their revised manuscript has not revised completely in accordance with the reviewers’ comments.  Therefore, there are still several points that the authors should revise their manuscript before the acceptance for “Antioxidants”.

  • In their answer to the comment of reviewer #2, the authors described that their research question is whether the effective constituents antrodin C isolated from the cultured mycelia of A. camphorate has the same ability of anticancer and signaling activation remains unclear. However, it seems to be a weak motive, and if so, the research question and their own answer/data are not indicated in their whole manuscript from the points of the view. Therefore, the authors should describe their fundamental research question for this investigation, as previously pointed out as “the authors should discuss both similarities and differences of the effects of antrodin C”. 

Response: We appreciate this remark of reviewer #2’s comment.

Several studies have demonstrated the inhibition of the antrodin C, a maleimide derivative isolated from A. camphorata mycelium in cancer progression and metastasis, including non-small cell lung cancer and breast cancer as well as head and neck squamous cell carcinoma [15-17]. Antrodin C potently inhibited the viability of human lung adenocarcinoma cell line, induced apoptosis triggered by ROS, and arrested the cell cycle at the G2/M phase by suppressing both the Akt/mTOR and AMPK signaling pathways. In addition, antrodin C was able to regulate epithelial-to-mesenchymal transition and metastasis of breast cancer cells via suppression of Smad2/3 and β-catenin signaling pathways. In our study, we demonstrated that the molecular mechanisms by which antrodin C-induced apoptosis and cell cycle G1 arrest in HCT-116 cells was associated with the activation of TNFα-mediated extrinsic and the mitochondria-mediated apoptotic cells death (Figure 3 and figure 6). However, antrodin C treatment increases the expression of TNFα through ROS-derived and ERK/AKT/p38 MAPK signaling pathways-mediated histone modification of H3K9K14ac will provide more insight into the mechanisms in anti-cancer in CRC.

These were revised in the discussion section, page 11-12, line 409-423.

2) Their answer to the comment 3 of reviewer #2 is insufficient, but their results in their answer to the comment 4 of reviewer #2, which demonstrated the importance of ROS production by antrodin C, is quite meaningful. Therefore, the authors should investigate whether antrodin C induce ROS production in normal epithelial cells HCoEpiC, and compare with that in colon cancer cells HCT-116.

Response: We appreciate this remark of reviewer #2’s comment and respond to the comment as follows:

Measurement of the intracellular ROS of ADC- treated cells for HCT-116 and HCoEpiC at 24 h was performed by H2DCFDA stain FACS analysis, while ROS production is expressed as the fold-change of the control group.

However, we have shown that antrodin C not significantly increased the generation of ROS in HCoEpiC cells with or without antrodin C treatment. HCT-116 cells treated with antrodin C increased intracellular ROS (Replay 1).
